# Enrichment of root-associated *Streptomyces* strains in response to drought is driven by diverse functional traits and does not predict beneficial effects on plant growth

Citlali Fonseca-Garcia[1,○], Dean Pettinga[1,○], Daniel Caddell[1,2], Hannah Ploemacher[1], Katherine Louie[3], Benjamin P. Bowen[3], Joelle Park[1,4], Jesus Sanchez[1], Alen Zimic-Sheen[1], Matthew F. Traxler[1], Trent R. Northen[3], Devin Coleman-Derr[1,2*]

**1** Plant and Microbial Biology Department, University of California at Berkeley, Berkeley, California, United States of America, **2** Plant Gene Expression Center, United States Department of Agriculture, ARS, Berkeley, California, United States of America, **3** The Joint Genome Institute, Lawrence Berkeley National Lab, Berkeley, California, United States of America, **4** Department of Plant Biology, University of California at Davis, Davis, California, United States of America

☯ These authors contributed equally to this work.
\* devin.coleman-derr@usda.gov

## Abstract

The genus *Streptomyces* has consistently been found enriched in drought-stressed plant root microbiomes, yet the ecological basis and functional variation underlying this enrichment at the strain and isolate level remain unclear. Using two 16S rRNA sequencing methods with different levels of taxonomic resolution, we confirmed drought-associated enrichment (DE) of *Streptomyces* in field-grown sorghum roots and identified five closely related but distinct amplicon sequence variants (ASVs) belonging to the genus with variable drought enrichment patterns. From a culture collection of sorghum root endophytes, we selected 12 *Streptomyces* isolates representing these ASVs for phenotypic and genomic characterization. Whole-genome sequencing revealed substantial variation in gene content, even among closely related isolates, and exometabolomic profiling showed distinct metabolic responses to media supplemented with drought- versus well-watered root tissue. Traits linked to drought survival, including osmotic stress tolerance, siderophore production, and carbon utilization, varied widely among isolates and were not phylogenetically conserved. Using a broader panel of 48 *Streptomyces*, we demonstrate that DE scores, determined through mono-association experiments in gnotobiotic sorghum systems, showed high variability and lacked correlation with plant growth promotion. Pangenome-wide association identified orthogroups involved in osmolyte transport (e.g., proP) and membrane biosynthesis (e.g., fabG) as positively associated with DE, though most associations lacked phylogenetic signal. Collectively, these results demonstrate that *Streptomyces* DE is not a conserved genus-level trait but is instead strain-specific and functionally heterogeneous. Furthermore, DE in the root

**Data availability statement:** All datasets and scripts for analysis are available through GitHub (https://github.com/deanpettinga/strep_manuscript) and Zenodo (https://doi.org/10.5281/zenodo.17554086). All the 16S rRNA short- and long-read data can be accessed through NCBI BioProject PRJNA655744. Raw metabolomics data is available through MassIVE (https://massive.ucsd.edu/) under accession: MSV000093246. Raw genomics data is also available through the JGI Genome Portal with the Proposal ID 502952.

**Funding:** This work was supported by US Department of Agriculture (CRIS 2030-21430-008-00D to D.C.-D.), USDA-NIFA (2019-67019-29306 to D.C.-D.), and the U.S. Department of Energy Joint Genome Institute (proposal 10.46936/10.25585/60001083 to D.C.-D.; https://ror.org/04xm1d337), a DOE Office of Science User Facility, supported by the Office of Science of the U.S. Department of Energy operated under contract DE-AC02-05CH11231. This is also a contribution of the Pacific Northwest National Laboratory (PNNL) Secure Biosystems Design Science Focus Area "Persistence Control of Engineered Functions in Complex Soil Microbiomes" (operated by the U.S. DOE under contract DE-AC05-76RL01830 to D.C.-D.). The funders had no role in study design, data collection and analysis, decision to publish, or preparation of the manuscript.

**Competing interests:** The authors have declared that no competing interests exist.

**Abbreviations:** ANI, average nucleotide identity; ASVs, amplicon sequence variants; CAS, Chrome Azurol S; CCS, Circular consensus sequences; COG, Clusters of Orthologous Groups; DE, drought enrichment; FAS, fatty acid synthase; HMMs, Hidden Markov Models; ONT, Oxford Nanopore Technologies; PBS, phosphate-buffered saline; RT, retention time; TSB, tryptic soy broth; TWYE, tap water–yeast extract.

microbiome was shown not to predict beneficial effects on plant growth. This work underscores the need to resolve functional traits at the strain level and highlights the complexity of microbe-host-environment interactions under abiotic stress.

## Introduction

Plants exist as holobionts, intimately associated with complex microbial communities that play critical roles in plant development, health, and stress tolerance [1]. These plant-associated microbiomes, particularly those colonizing the rhizosphere and root endosphere, can influence a wide range of plant phenotypes including nutrient acquisition, pathogen resistance, and abiotic stress resilience [2,3]. Under environmental stress, such as drought, plants undergo shifts in root exudation and physiology that reshape their microbiomes in ways that have been proposed to buffer the host against damage [4,5]. Understanding the composition and functional traits of microbial taxa that associate with plant roots under stress conditions is essential for improving crop resilience through microbiome-informed strategies [6,7].

One group of microbes repeatedly found to increase in relative abundance in the roots of drought-stressed plants is the Actinobacteria, particularly members of the genus *Streptomyces* [5,8–10]. Levels of *Streptomyces* increased in the root microbiome across a wide range of more than 30 angiosperm plant hosts, and overall levels of *Streptomyces* abundance correlated with positive host phenotypic outcomes during drought stress [9]. Some recent studies have also shown that bioinoculation with specific *Streptomyces* species in an agricultural setting mitigated drought stress, potentially through enhancing antioxidant enzymes and optimizing osmotic regulation [11]. Collectively, these prior studies demonstrate that the genus *Streptomyces* as a whole exhibits drought enrichment (DE), and may be an important source of plant growth promotion during drought.

While *Streptomyces* enrichment under drought is well-documented, less is known about the specific functional diversity within this genus that may explain its ecological success in the rhizosphere and its variable effects on plant hosts. Evidence from prior studies suggested several potential causes for this DE: *Streptomyces* form spores, enabling survival during harsh conditions, they exhibit high osmotic stress tolerance, produce siderophores, and metabolize diverse plant-derived carbon substrates [12,13] *Streptomyces* species are renowned for their metabolic versatility, including the production of antibiotics, siderophores, osmoprotectants, and plant hormones, the presence and expression of these traits can vary dramatically even among closely related strains [14–16]. In drought-stressed rhizospheres, these functional differences could influence microbial fitness, competitive interactions, and ultimately, the degree of benefit conferred to the plant. Given that the genus *Streptomyces* is well-known to be fast-evolving [17–19], we hypothesized that *Streptomyces* DE in sorghum roots is not a uniform, lineage-level phenomenon, but instead driven by isolate-specific traits varying across a genetically diverse pool of endophytic strains. Furthermore, we propose that DE and its underlying causes—such as osmotic stress tolerance,

secondary metabolite production, siderophore production, and utilization of drought-associated root exudates—could be uncoupled from the strains' capacity to promote plant growth.

To test these hypotheses, we began by performing a metagenomic analysis of the impact of drought on the sorghum root microbiome using both V3-V4 and full-length 16S rRNA amplicon sequencing, with an emphasis on understanding genotypic variation in *Streptomyces* and its relationship to DE. Using these data, we identified a collection of closely related *Streptomyces* root endophyte isolates obtained from *Sorghum bicolor*, an important feedstock and bioenergy crop [20], and subjected them to genomic, exometabolomic, and phenotypic characterization. We demonstrated that the single most dominant *Streptomyces* amplicon sequence variant (ASV) observable through V3-V4 16S rRNA sequencing in the sorghum root represents a wide diversity of isolates with substantial variation in genomic content and traits purported to be associated with the DE phenomenon. Through pangenomic analysis, we showed that both DE and plant growth promotion vary widely among this collection of isolates, and that neither trait correlates with the other or the underlying phylogenetic distance between them, underlining the importance of considering isolate-level genomic and phenotypic variation when exploring and predicting crop-microbe interactions.

## Results

The genus *Streptomyces* has been repeatedly exhibited strong increased relative abundance under drought, relative to irrigated conditions (referred to as DE) through a variety of metagenomics sequencing techniques [5,8–10], but the degree to which this DE holds for individual root-associated *Streptomyces* remains unclear. To explore this, we performed a field experiment in which the root microbiomes of drought-stressed and control-irrigated sorghum were sampled and analyzed via 16S rRNA sequencing using both V3-V4 (Illumina) and full-length (PacBio) primer sets (Fig 1A). Interestingly, Actinobacterial amplification was lower in both conditions when the full-length 16S rRNA primers were used (S1 Table). *In silico* analysis of Actinobacteria, Proteobacteria, Bacteroidetes, and Firmicutes sequences in the reference database used for taxonomic classification revealed that a much smaller fraction of Actinobacterial sequences were "amplifiable" with the PacBio full-length primer set than with the V3-V4 primer set, potentially accounting for the discrepancy (S2 Table); numerous studies have demonstrated that use of different 16S rRNA primer sets leads to significant and reproducible differences in the resulting taxonomic profiles [21–23]. Importantly, however, both the V3-V4 and the full-length 16S datasets demonstrated significant Actinobacterial DE in the roots of drought-stressed sorghum. Further analysis of the V3-V4 16S rRNA dataset revealed a single dominant *Streptomyces* ASV (ASV650) in the root endosphere of sorghum as the most abundant (Fig 1B); it represented 12.3% of the total relative abundance within the roots of watered plants and 16.7% ($P = 0.017$, S1 Table) within drought-stressed roots. A comparison of the V3-V4 and full-length 16S rRNA ASVs reveals that ASV650 shares 100% nucleotide identity across the V3-V4 region with five distinct ASVs detected using the full-length 16S primer sets (Fig 1C).

An exploration of the enrichment patterns of these five full-length ASVs (referred to as AC2 through AC36) demonstrated that all exhibit some level of DE in the root microbiome, although the level of enrichment varied (S1 Table). Only one of these ASVs (AC2) was detectable in roots of both drought and control treatments (watered sorghum); three of the other ASVs were only detected following drought stress, but are observed in multiple samples, with an average relative abundance varying between 1.94% and 3.15% of the total root microbiome readcounts. The final ASV, AC36 was only detected in a single drought-stressed root sample, and accounted for only 0.011% of total relative abundance. When combined, these five ASVs explained 1.39% of total relative abundance under watered conditions, and 9.68% under drought conditions, corroborating the pattern of enrichment observed for ASV650. Collectively, these data revealed that while both 16S rRNA sequencing approaches exhibit a general trend of DE for *Streptomyces* in the root microbiome, levels of enrichment differed between ASVs in the full-length 16S rRNA dataset.

As full-length 16S rRNA sequencing revealed additional diversity in *Streptomyces* present within the sorghum root not observable via shorter amplicon sequencing, we hypothesized that root-associated *Streptomyces* isolates matching

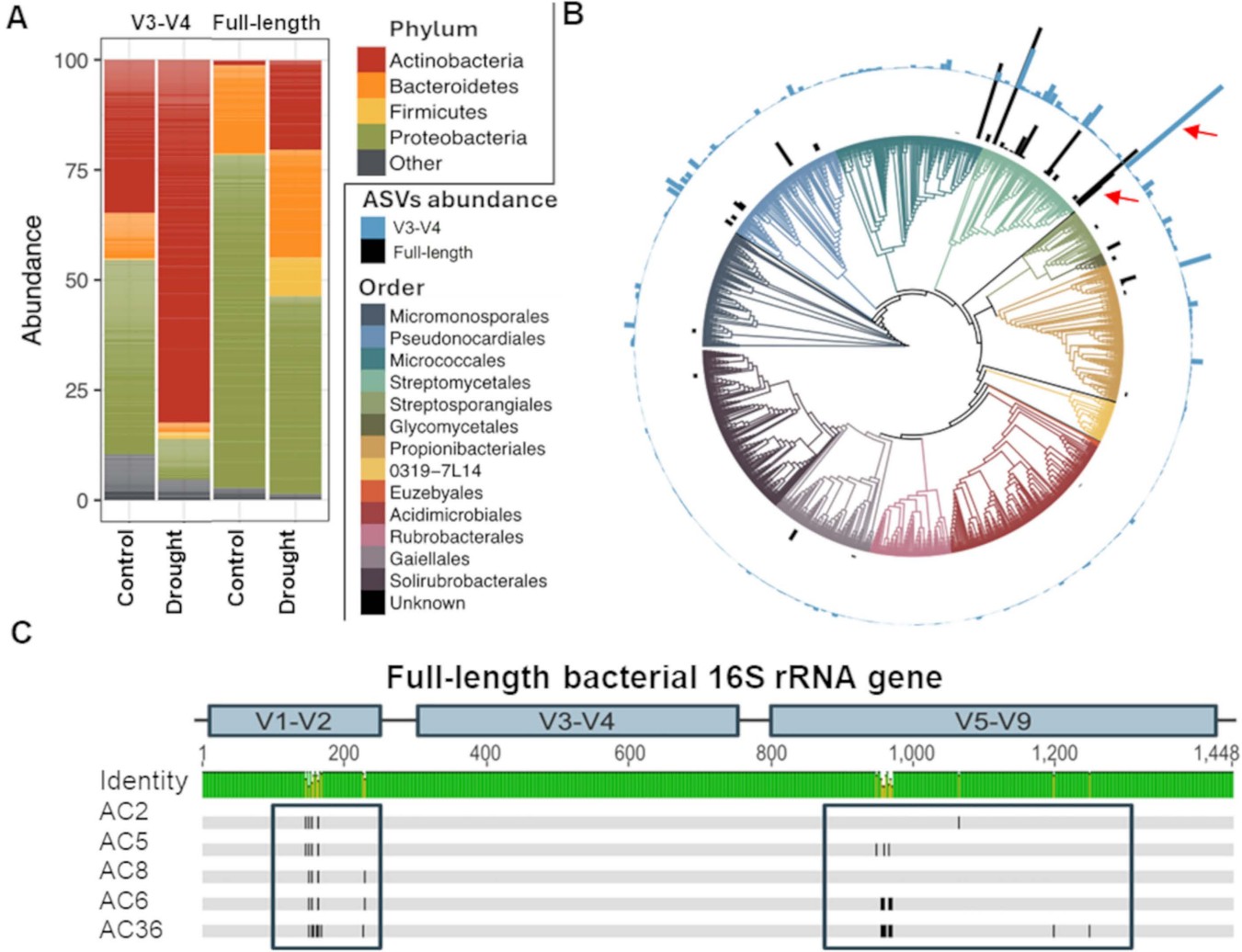

**Fig 1. High-resolution taxonomic profiling reveals hidden diversity among root endophytic *Streptomyces* under drought and control conditions. (A)** 16S rRNA amplicon sequencing of the root endosphere under drought and control conditions using both V3-V4 and full-length 16S rRNA primer sets. **(B)** Phylogenetic tree of all the Actinobacteria ASVs identified by V3-V4 16S rRNA amplicon sequencing (outer ring) or full-length 16S rRNA amplicon sequencing (inner ring) of the root endosphere of field-grown sorghum at order level. The red arrows highlight the single dominant *Streptomyces* ASV identified via V3-V4 and the five corresponding ASVs identified via full-length sequencing. The tree shown here represents topology only (branch lengths not to scale) and is intended to illustrate taxonomic relationships rather than evolutionary distances. **(C)** Plot of single nucleotide polymorphism (SNP) locations across the full-length 16S rRNA region of the five ASVs identified via PacBio sequencing, reveals 100% sequence conservation within the V3-V4 region and additional genotypic diversity elsewhere within the 16S rRNA gene. The data underlying this Figure can be found in https://doi.org/10.5281/zenodo.17554086.

these ASVs may reveal further diversity in response and allow for deeper functional and physiological characterization of their growth in the rhizosphere. To enable this, we explored a previously described sorghum root endophyte isolate collection developed in our lab [24]; a set of 170 *Streptomyces* from this collection were genotyped via full-length 16S rRNA sequencing and screened for 100% nucleotide identity homology matches to the dominant ASV650 observed in our original V3-V4 dataset. In total, we identified 28 isolates from the collection that fit this criteria, representing exact matches to four of the five AC ASVs identified by full-length 16S rRNA sequencing (isolates matching AC36 were not identified). Visual inspection of all 28 *Streptomyces* isolates grown on spore induction media revealed large variation in pigmentation

phenotypes (Fig 2A), suggesting additional layers of metabolic and/or physiological variation could be present within each AC group.

Next, from each group of isolates matching the four AC ASVs (referred to as AC groups), we selected three isolates for exometabolomic characterization following growth on three distinct media types (Fig 2A, total strains $n = 12$) versus uninoculated control media. Data from minimal media (TWYE) revealed clustering by AC group for isolates belonging to AC2 and AC5, with less distinction between isolates belonging to AC6 and AC8, (Fig 2B and Fig A in S1 Text). Additionally, these data suggest that metabolite profiles for some individual isolates (e.g., DC14) are different from others within their AC group. To test how growth on more rhizosphere-relevant carbon substrates might impact these results, all 12 isolates were next profiled using exometabolomics following growth on media containing ground sorghum root tissue from either drought-stressed or control-irrigated plants. These data demonstrate that isolates fed drought-stressed root tissue have distinct metabolic profiles from those fed control root tissue, and lend further support to the previous observation that isolates belonging to group AC2 and AC5 have distinct exometabolomic profiles from the other strains (Fig B in S1 Text).

Next, we hypothesized that the observed differences in metabolic profiles among these 12 isolates may correlate with differences in other phenotypes, including those hypothesized to be associated with DE in the root environment. To test this, we assessed variation among the 12 isolates for several microbial phenotypes with proposed links to the observed DE of Actinobacteria including: general osmotic stress resistance [25], iron acquisition ability [5,26], and carbon resource utilization [24]. Importantly, the phenotypes showed strong between-isolate variability (S3 Table). This included single-locus traits, such as levels of the individual bacterial siderophore Ferrioxamine E-F (measured through exometabolomics, Figs C, D in S1 Text), as well as more complex phenotypic traits likely explained by multiple loci, such as osmotic stress tolerance and total siderophore production (Figs E, F in S1 Text).

To assess whether phenotypic variation between these 12 isolates correlated with phylogenetic relatedness, whole genome sequencing of the 12 strains was performed using a combination of PacBio and Illumina sequencing, and both pangenomic and phenotypic information was plotted in Anvi'o (Fig 3) [27]. An exploration of the genomic data supports the

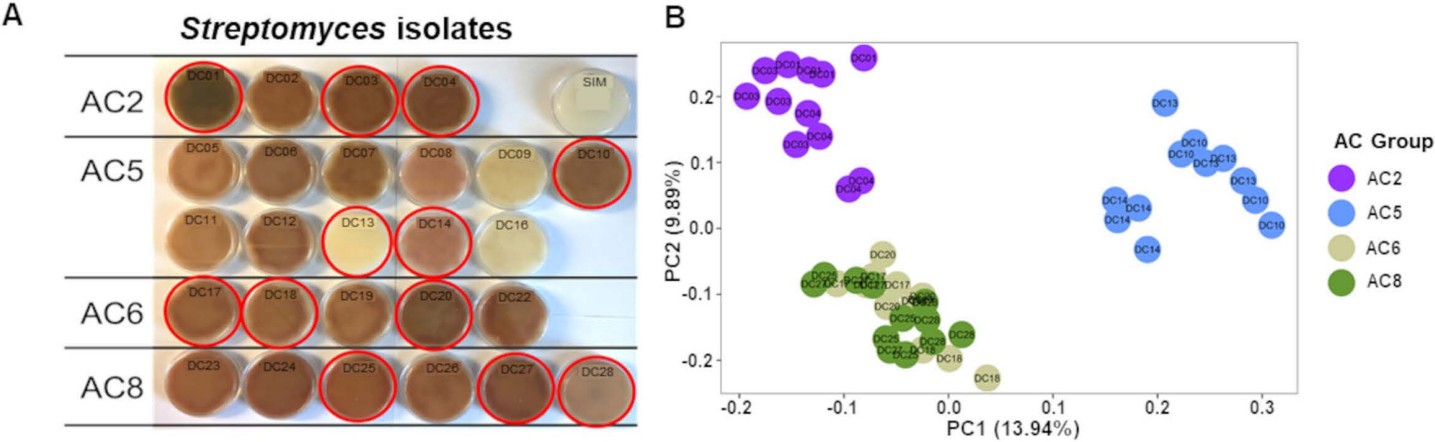

**Fig 2. Phenotypic and metabolomic differentiation among *Streptomyces* isolates sharing a dominant root-derived V3-V4 16S rRNA genotype.**
**(A)** Image of the growth (on spore induction media, or SIM) for all 28 *Streptomyces* isolates with 100% identity to the dominant ASV identified in the root via V3-V4 16S rRNA sequencing. Isolates are arranged by the AC group (full-length 16S rRNA genotype) that they belong to, highlighting within-group differences in pigmentation patterns. Isolates selected for additional metabolomic, genomic, and phenotypic characterization ($n = 12$) are indicated by red circles around the corresponding petri dish. **(B)** Ordination of polar exometabolomic profiling of the 12 strains following growth on liquid tap water–yeast extract (TWYE) medium showing distinct clustering of AC group 2 and 5 (replication $n = 4$). AC groups are presented by colors: AC2, purple; AC5, light blue; AC6, green mist; AC8, green. *Streptomyces* isolate IDs are indicated in the corresponding plate. The data underlying this Figure can be found in https://doi.org/10.5281/zenodo.17554086.

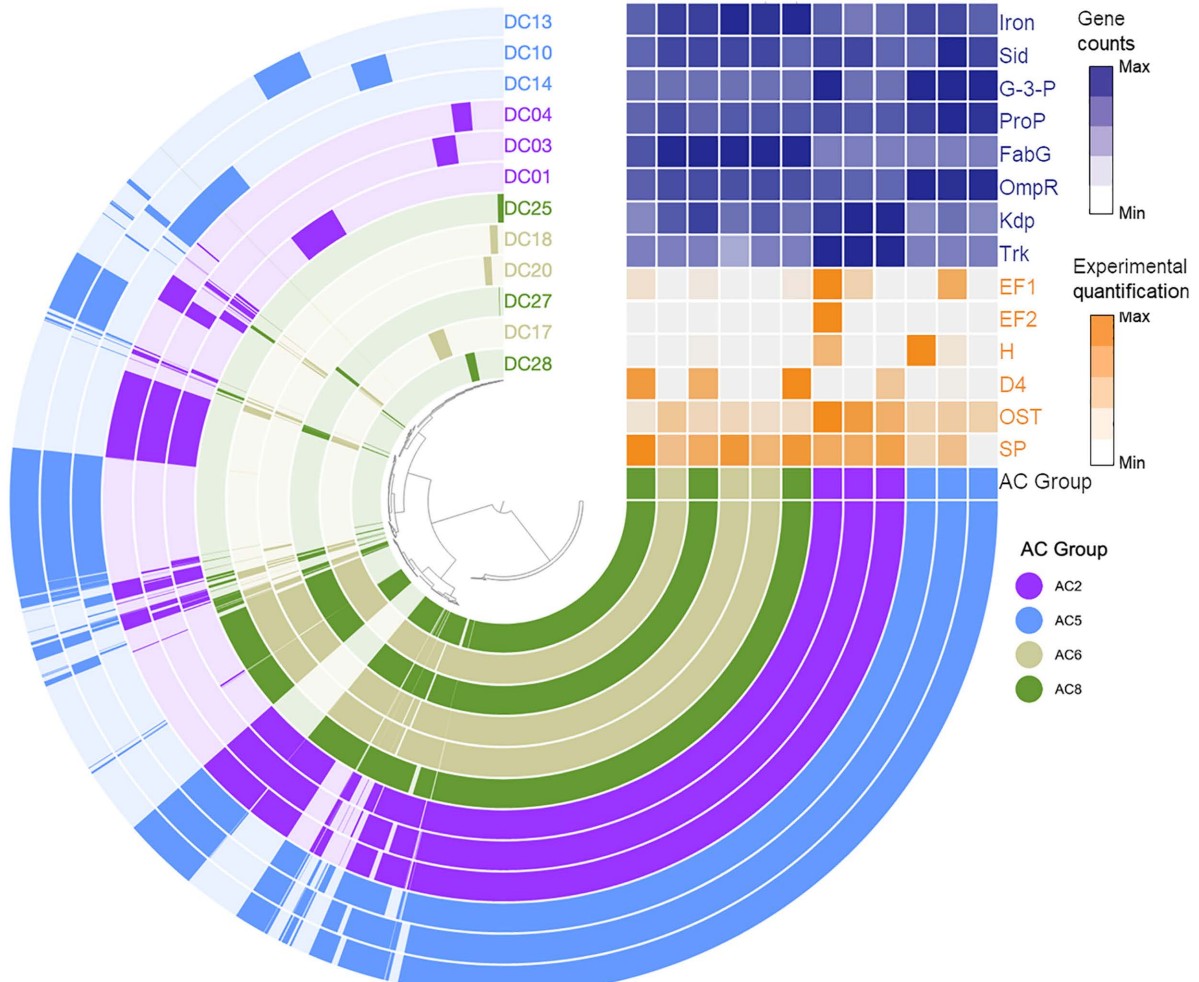

**Fig 3. Pangenomic visualization of the genomic organization of all 12 *Streptomyces* isolates representing each of the four AC groups.** Hierarchical clustering is based on orthogroup presence/absence using Euclidean distance and Ward's linkage method. Colored rings show orthogroup presence/absence per isolate, the outer ring shows the number of contributing genomes, and the second outer ring shows the number of genes present in each orthogroup. The metadata table shows 1) experimental phenotype data (orange) of total siderophore production (SP, halo size on CAS-LB media), growth under osmotic stress (OST, 1.0M sorbitol), and the relative production of four individual siderophores in response to media supplemented with drought- vs. well-watered root tissue (EF1 and EF2, ferrioxamine EF; H, O-methyl desferrioxamine H, D4, derrioxamine D4), and 2) genome analysis data (dark blue), showing the number of gene copies that match with specific COG function search terms related to iron acquisition (iron and Sid, siderophores), carbon resource utilization (G-3-P, glycerol-3-phosphate), and osmotic stress response (ProP, FabG, OmpR, Kdp, Trk). Scale values represent Z-scores, where darker blue indicates more gene copies, and darker orange indicates more siderophore production, more growth under osmotic stress, or a larger halo. The data underlying this Figure can be found in https://doi.org/10.5281/zenodo.17554086.

general topology of the phylogeny established through full-length 16S rRNA sequencing, with most members belonging to a given AC group sharing greater genomic homology with other members of the group than those outside the group. Of note, we observed the greatest differences in genomic content and organization between clusters AC2 and AC5, while isolates belonging to AC6 and AC8 appeared more similar to one another. Next, the genomes for each strain were screened for copy number variation in genes putatively associated with DE, including iron acquisition, carbon resource utilization, and osmotic stress response (Fig 3). Genes associated with osmotic stress tolerance included *ProP*, which encodes an osmoregulatory proton symporter [28]; *OmpR*, a transcription factor regulating the production of outer membrane porins

[29]; *Kdp*, member of an ATP-dependent potassium ion uptake system [30]; *Trk*, a potassium ion transporter [31]; and *FabG*, an essential reductase in the fatty acid biosynthesis pathway [32]. Importantly, for both the genomically-derived phenotypes (Fig 3, in blue) and the previously lab-quantified phenotypes (Fig 3, in orange) we found evidence of both inter- and intra-AC group variation. Collectively, these data suggest that phylogenetic relatedness among isolates does not predictsimilarity for DE-associated phenotypes.

Based on these results, we hypothesized that *Streptomyces* DE within the root environment may follow a similar pattern of trait variability across the genus [9,10,24]. To test this, we selected a larger cohort of 48 *Streptomyces* from our sorghum root-endophyte isolate collection for genomic characterization and DE phenotyping. This new cohort included all 12 isolates analyzed previously, plus an additional 36 isolates from our collection to maximize taxonomic breadth (based on V3-V4 16S rRNA sequence data). To provide an accurate estimate of phylogenetic relatedness among isolates, whole genome sequencing was performed (Fig 4); these data revealed 17 unique species (ANI > 95%) and 21 unique strains (ANI > 99%) across the cohort. To estimate functional diversity across the cohort, we estimated the core (*n* = 2,123) and pangenome (*n* = 17,557). To illustrate the heterogeneity of gene content at different taxonomic classifications, we compared the numbers of shared and unshared orthogroups—a set of genes descended from a single gene in the last common ancestor of the considered genomes [33] between genomes within the genus, species, and strain level. Even within

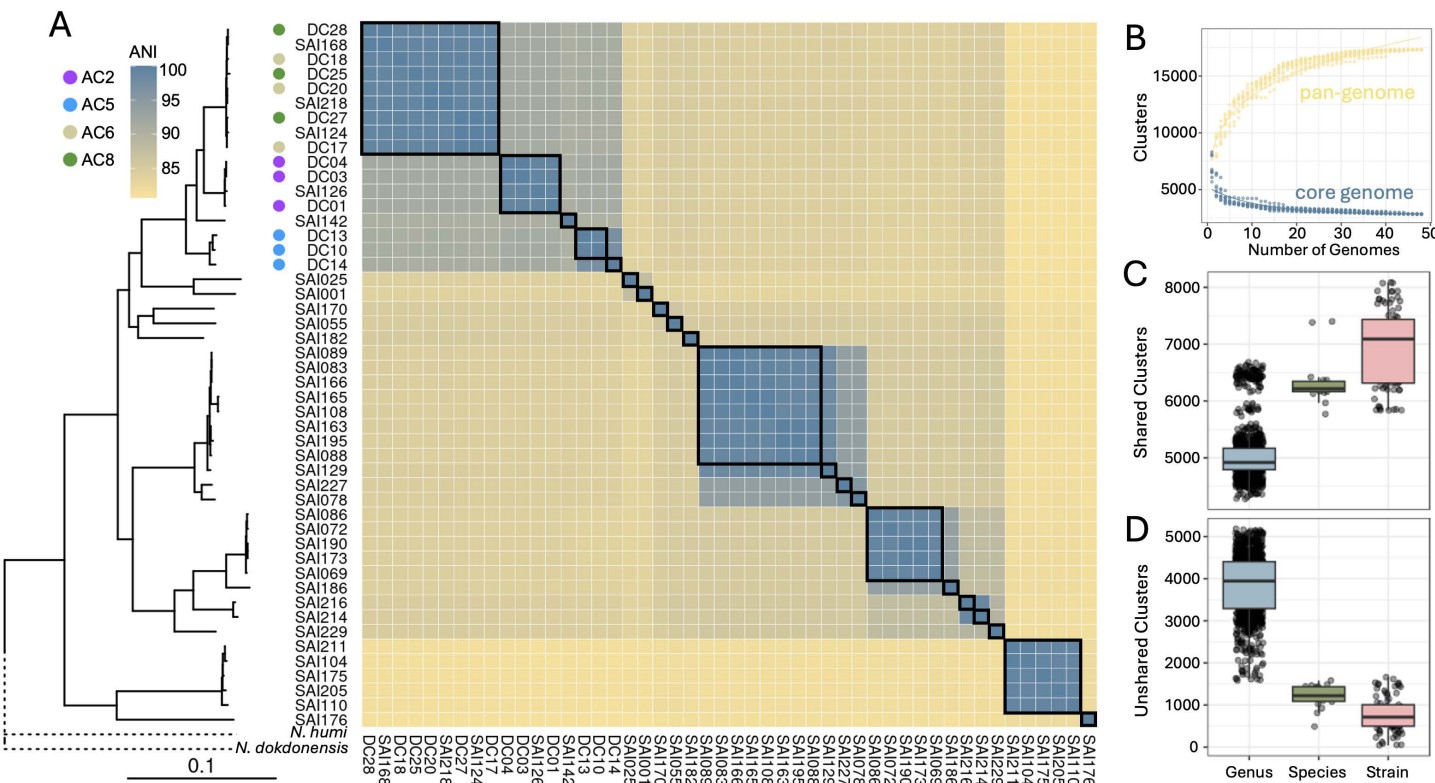

**Fig 4. Phylogenomic and comparative genomic analysis of 48 *Streptomyces* isolates.** (A) Phylogenetic tree of all 48 *Streptomyces* isolates based on alignment of 138 single-copy genes and rooted by a *Nocardioides* outgroup. Relevant isolates are annotated with AC Group. Corresponding ANI heatmap aligned to tree tips shows levels of relatedness among isolates calculated with *fastANI*. Black boxes within the heatmap indicate unique strains (>99% ANI). (B) Pan- and core genome rarefaction simulations fit to power law and exponential decay curves, respectively using *pagoo* R package. (C) Shared and (D) unshared orthogroup counts between pairs of isolates compared at different taxonomic levels across the collection. Orthogroups identified by *Orthofinder* and taxonomic classification determined by pairwise ANI (Genus: <95%, Species: 95%–99%, Strain: >99%). Scale bar represents the mean number of substitutions per site. The data underlying this Figure can be found in https://doi.org/10.5281/zenodo.17554086.

strains, we observed high variability in shared gene content from a minimum of 5,835–8,085 orthogroups, and as many as 1,660 unshared orthogroups. Genome fluidity comparisons—a measure of gene-level dissimilarity across the pangenome [34]—within species ranged from 0.04 to 0.12 and within strain from 0.03 to 0.06, suggesting that differentiation by ANI underestimates functional differentiation. Collectively, the presented pangenome captured high levels of genetic diversity, even within closely related *Streptomyces* strains isolated from a single field site.

Next, to assess DE scores for this cohort, each individual *Streptomyces* isolate ($n = 48$) was grown in mono-association with young sorghum seedlings in sterilized calcined clay under either watered or drought conditions for a period of 4 weeks, at which point plants were harvested and root colonization for the isolate was assayed by qPCR using Actinobacteria-specific primers. A single DE score was calculated for each isolate by averaging replicates ($n = 4$). In support of our hypothesis, we observed high DE variance (1.16) across the 48 isolates with scores ranging from −2.48 (depleted) to 1.73 (enriched) (Fig 5A). The observation that many tested *Streptomyces* isolates exhibit depletion under drought when grown with sorghum roots lacking a normal rhizosphere microbiome stands in contrast to many observations made in native field contexts [9,10,24], and demonstrates that individually assessed *Streptomyces* phenotypes may vary considerably for this reportedly conserved trait.

We next hypothesized that the combination of complete genomes and variable DE scores for this cohort of related and environmentally co-habitating *Streptomyces* may enable us to identify genetic modules positively correlated with increased DE in the root. To identify pangenomic variation contributing to the observed variation in DE, orthogroup copy number variation was assessed across the pangenome. Orthogroups with suitable variance (>0.2) in copy number were tested for association with the DE phenotype using phylogenetic regression. For each orthogroup, a linear regression for DE as a function of orthogroup copy number was fit while simultaneously estimating the degree to which phylogenetic signal (Pagel's $\lambda$) explains variation among the model residuals. Among analyzed orthogroups, only 4 out of 2,735 exhibited $\lambda > 0$. In total, 342 clusters passed our filtering thresholds for robust regression results including only a single cluster with any degree of phylogenetic signal estimated in its model. These results suggest an overwhelming lack of phylogenetic signal among individual orthogroup copy number variation correlating with DE.

Next, results were tested for functional enrichments among the top DE-associated orthogroups. Significant results included two categories: *inorganic ion transport and metabolism* and *lipid transport and metabolism.* Across both of these high-level categories were two notable osmolyte-related orthogroups (Fig 5B). First, within the *Lipid transport and metabolism* Clusters of Orthologous Groups (COG) category, we detected multiple instances of *FabG* (COG1028), an enzyme essential for fatty acid biosynthesis. In total, 8 distinct orthogroups annotated as *FabG* displayed strong association between copy number and DE (Fig 5C). Second*,* we also detected association between gene copy number and DE in 7 independent orthogroups for *ProP* (COG0477), a proton symporter necessary for osmolyte transport (Fig 5D). Taken together, these results suggest that biosynthetic machinery and transporters for root and rhizosphere-enriched osmolytes may increase bacterial fitness under drought in the root environment.

Finally, it has been proposed that the phenomena of *Streptomyces* DE in the root may support host growth and stress tolerance, and individual isolates of *Streptomyces* have been shown to confer such benefits to their hosts [24]. Given the high variability in DE responses observed across our 48 isolates, we next sought to determine if the observed degree of DE correlated with beneficial impact on host performance across this cohort of isolates. To test this, we analyzed shoot biomass data for each plant grown in the previous DE experiment (Fig 6). Importantly, we observed no evidence for correlation between plant growth promotion and DE (water: Spearman's $\rho = 0.16$ $P = 0.26$; drought: Spearman's $\rho = −0.2$, $P = 0.17$) or plant drought tolerance promotion with DE (Spearman's $\rho = 0.00$ $P = 0.99$). Examination of the phylogenomic context of both DE and plant growth promotion demonstrated that neither trait appeared phylogenetically conserved among closely related isolates. For example, divergent DE was noted between sister isolates DC01 and SAI126. These isolates were among the highest and lowest DE performers, respectively. Interestingly, these strains also had strongly contrasting patterns of plant host growth promotion under watered conditions and drought tolerance promotion. Notably,

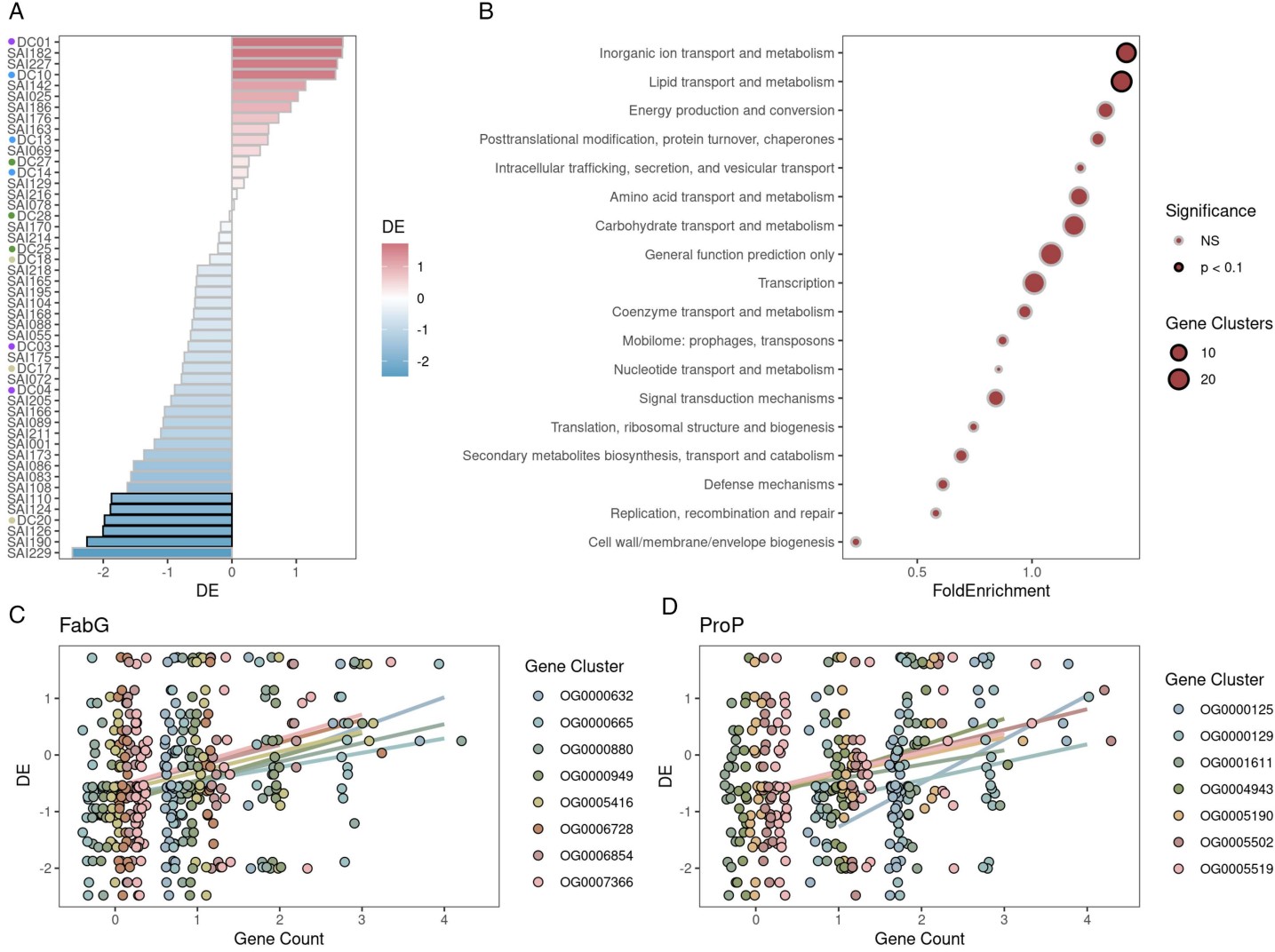

**Fig 5. Drought enrichment of *Streptomyces* isolates and associated gene function reveal candidate mechanisms for root colonization under drought stress. (A)** Drought enrichment (DE) scores for all 48 *Streptomyces* isolates measured by lineage-specific 16S rRNA qPCR amplification of bacterial DNA from sorghum root extracts after 4 weeks of growth under drought or irrigation treatment in gnotobiotic microbox containers. DE, $log_2$(Drought/Water) where "Drought" refers to the mean abundance of 4 replicates under drought and "Water" refers to mean abundance of 4 irrigated replicates. Red bars represent average drought enrichment and blue bars represent depletion. **(B)** Orthogroups with significant, positive DE-copy number associations were tested for COG term enrichment relative to all analyzed orthogroups by hypergeometric test. Among the most significantly enriched COG terms *Lipid transport and metabolism*, and *Inorganic ion transport and metabolism* were multiple orthogroups with **(C)** essential fatty acid biosynthesis gene *FabG* ($n = 8$) and **(D)** osmolyte-proton symporter *ProP* ($n = 7$) annotations, respectively. The data underlying this Figure can be found in https://doi.org/10.5281/zenodo.17554086.

SAI126 treatment—which exhibited negative DE—reduced plant biomass under watered conditions, yet positively impacted plant biomass and water content under drought. In contrast, DC01—which exhibited positive DE—increased plant biomass under watered and droughted conditions, but reduced water content under drought. Such variation among closely related bacteria suggests a lack of phylogenetic signal for DE, plant growth promotion, and plant drought tolerance promotion. This was confirmed with formal tests for two estimates of the effect of phylogeny upon DE trait variation (Fig 6) (Pagel's $\lambda = 7.3 * 10^{-5}$, $P = 1$; Blomberg's $K = 2.6 * 10^{-6}$, $P = 0.35$). Similar results were obtained in tests of the effect of

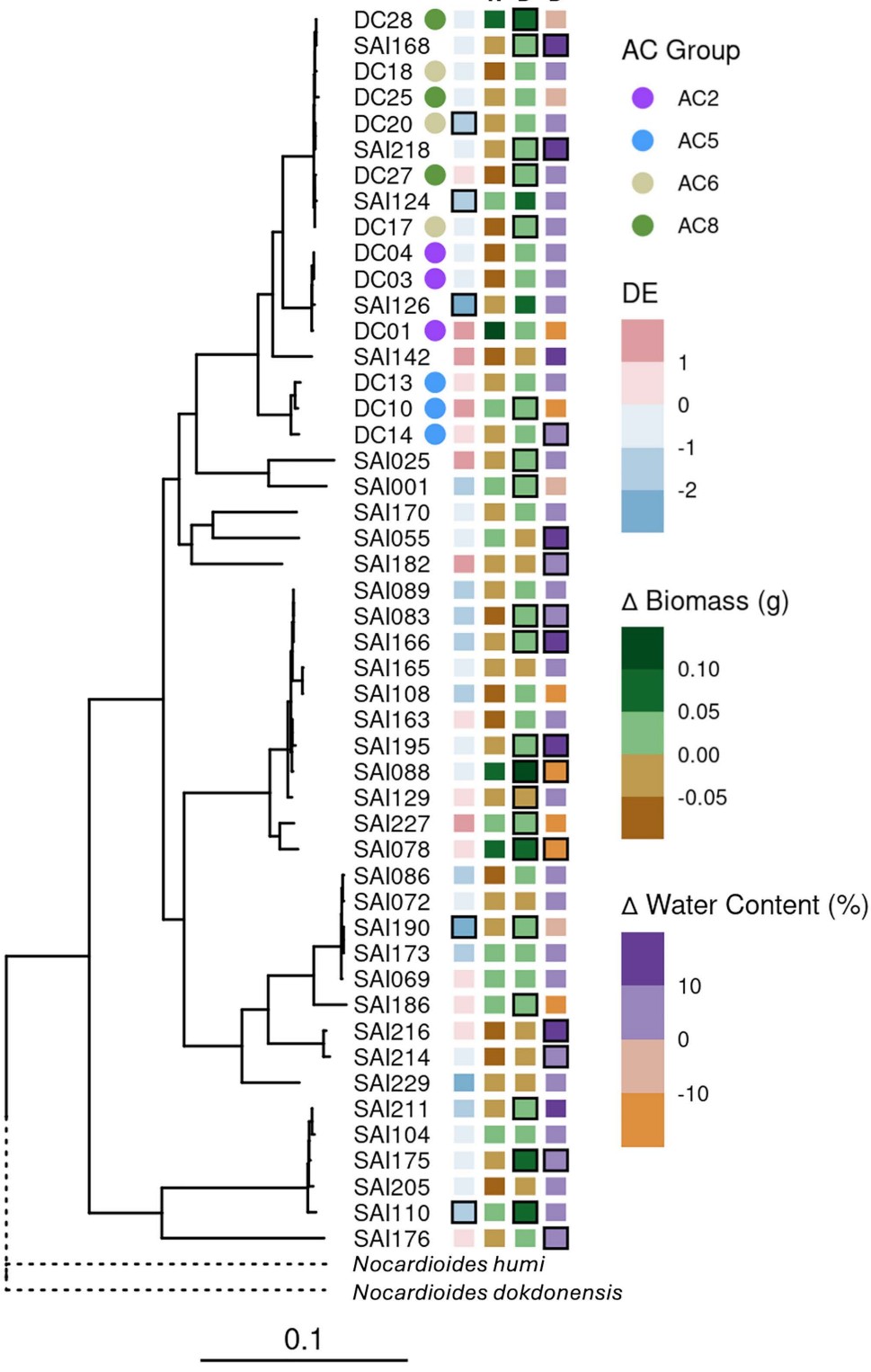

**Fig 6. Phylogenomic analysis reveals that *Streptomyces*-mediated drought enrichment and plant growth promotion are variable and not phy-logenetically conserved.** Phylogenetic tree of all 48 *Streptomyces* isolates based on alignment of 138 single copy genes and rooted by a *Nocardioides* outgroup. Corresponding annotation columns at tree tips include AC Group, DE, Δ Biomass (g) under watered conditions, Δ Biomass (g) under drought

conditions, and Δ Water Content (%) under drought conditions. "DE" refers to enrichment of the isolate in sorghum roots in drought relative to watered conditions in a mono-association microbox experiment: DE = $\log_2$(Drought/Water) where "Drought" refers to the mean abundance of 4 replicates under drought and "Water" refers to mean abundance of 4 irrigated replicates as measured by lineage-specific qPCR. Isolates with significant mean differences (FDR < 0.05) between "Drought" and "Water" abundances by t test and adjusted for multiple testing with Benjamini-Hochberg method are outlined in black. Δ Biomass (g) represents the mean difference in dry shoot biomass under Water – column "W" – and Drought – column "D" – conditions, respectively, relative to mock-treated control plants. The final column represents the mean difference in % Water Content of shoot tissue under drought conditions relative to mock-treated control plants. Tiles outlined in black represent significant differences (FDR < 0.05) relative to mock-treated plants by t test and adjusted for multiple testing with Benjamini–Hochberg method. Scale bar represents the mean number of substitutions per site. The data underlying this Figure can be found in https://doi.org/10.5281/zenodo.17554086.

phylogeny on plant growth promotion under watered conditions—Pagel's $\lambda = 7.3 * 10^{-5}$, $P = 1$; Blomberg's $K = 3.2 * 10^{-6}$, $P = 0.35$—and drought—Pagel's $\lambda = 7.3 * 10^{-5}$, $P = 1$; Blomberg's $K = 8.5 * 10^{-6}$, $P = 0.12$. Lastly, we found no effect of phylogeny on the impact of percent water content of the plant host under drought—Pagel's $\lambda = 7.3 * 10^{-5}$, $P = 1$; Blomberg's $K = 6.1 * 10^{-6}$, $P = 0.3$. Collectively, these data demonstrate a lack of connection between the ability to confer growth benefit or drought tolerance to the host, the degree of DE, and the phylogenetic context of the isolate.

## Discussion

Our study provides a high-resolution genomic and phenotypic characterization of root-associated *Streptomyces* diversity and highlights a key limitation of using amplicon sequencing based characterization of the microbiome as the sole means of inquiry. We demonstrate that both traditional and current gold standard 16S rRNA sequencing approaches (variable-region and full-length, respectively) fail to fully capture the *Streptomyces* genetic variation present within the rhizosphere, obscuring important phenotypic and functional differences in the genus that are relevant to its survival in this environment and its impact on host physiology. In other studies, 16S rRNA markers have been found insufficient to infer metabolic differentiation among *Streptomyces* strains [35,36], potentially because loci responsible for secondary metabolite production are subject to lateral gene transfer [37], resulting in rapid evolution. It has also been observed that isolates of *Streptomyces* with 100% 16S rRNA nucleotide identity can have as low as 84.42% ANI [38], suggesting that a single ASV could theoretically capture the average abundance over the entire genus in a 16S dataset. The data in our study also suggest that even full-length 16S rRNA ASV-based phylogenies of root-associated *Streptomyces* are inconsistent with genome-wide ANI-based taxonomic assignments, something that has also recently been observed in a broad survey of publicly available genomes belonging to this genus [39]. These issues with 16S rRNA resolution are not unique to *Streptomyces* and have been reported in diverse microbiome studies from many fields. In medicine, 16S rRNA sequencing resolution fails to distinguish between pathogenic and non-pathogenic isolates of genera commonly identified in clinical samples, including *Bacillus* and *Anaplasma* [40–42]. In marine biology, the most abundant clades are widely distributed across the world, and although ecotypes are closely related, they display distinct niche adaptations to various parameters such as temperature, salinity, and hydrostatic pressure [43]. For example, the *Synechococcus* clades III and IV thrive in distinct oceanic conditions, but could not be resolved based on the V6 hypervariable region of the SSU rRNA gene [44]. In contrast, despite high phylogenetic variability, functional composition can be highly similar, asshown in bacterial communities associated with macroalga *Ulva australis* [45]. Collectively, these observations highlight the importance of integrating high-resolution, genome-resolved approaches, such as whole-genome sequencing or metagenomics, alongside 16S rRNA profiling to more accurately assess microbial diversity, resolve functionally relevant taxa, and better understand the ecological and physiological roles of microbes within complex communities.

In this study, we assessed *Streptomyces* DE at both the isolate and metagenome scale. Based on our field data and many other field studies of droughted rhizospheres [8,10,24,46], we expected consistent, positive DE among the *Streptomyces* isolates tested in this study. In fact, only 16 of 48 isolates displayed positive DE scores when grown in

mono-association with host sorghum. This discrepancy is likely explained by several key differences in the experimental design of the prior and current studies. First, the highly controlled and artificial abiotic environments of the gnotobiotic experiments described above likely differ from the field conditions wherein *Streptomyces* enrichment has been routinely observed. These differences include the complexity and carbon content of the soil substrate, soil pH and nutrient availability, diurnal temperature fluctuations, and the duration and intensity of drought [24]. Second, and perhaps more importantly, the microbial composition of the rhizosphere in our lab-based experiment differs dramatically from field studies. In the present study, we applied each isolate to sorghum seedlings in mono-association under gnotobiotic conditions such that no competition or coordination with other microbes (bacteria or fungi) occurs. It is therefore feasible that the observed *Streptomyces* DE in native contexts requires the presence of other microbes. For example, an increase in *Streptomyces* abundance during drought may require a direct or indirect (host-mediated) biological interaction with other members of the rhizosphere microbiome. Recent work has shown that some strains of *Streptomyces* can respond to chemical signals and other microbes in their environments to alter growth habit and increase rates of expansion [47,48]. Alternatively, observed increases in *Streptomyces* abundance in the field may result from the fact that 16S rRNA amplicon sequencing in a community context constrains the sum of all microbes quantified such that a decrease in abundance of one microbe necessitates the increase in the abundance of all others [49,50]. Indeed, in a prior field-based study [24], a quantification of phylum-level absolute abundance revealed that Actinobacteria collectively exhibit an absolute decrease in abundance following severe drought, but show significantly greater resilience to this stress than three other phyla tested (Proteobacteria, Firmicutes, and Bacteroidetes). Taken together, these observations suggest that the apparent enrichment of *Streptomyces* under drought observed in field studies may reflect a combination of relative abundance shifts due to the depletion of more drought-sensitive taxa and context-dependent ecological interactions that cannot be replicated in isolation, underscoring the importance of studying microbial dynamics within their native community and environmental frameworks.

Our study also explored the extent to which DE is shared among closely related *Streptomyces*, revealing a lack of phylogenetic conservation in this trait. Such incongruence of phenotype and genotype has been observed with other traits in *Streptomyces* [51]. This phenomenon may be explained by distinct genomic characteristics of *Streptomyces*. Evolutionary studies in the genus suggest that extremely high recombination rates exceed the impact of mutation, decays clonality, and challenges traditional tree shaped phylogenies [52–54]. Such recombination events have been documented at both small and large timescales [54,55] with major implications for evolution within the genus. In addition, extensive horizontal gene transfer via integrative and conjugative elements has been documented among isolates derived from a millimeter-scale rhizosphere population [37,56]. This highly dynamic transfer among closely related isolates has the potential to impact fitness—both positive and negative—and facilitate adaptation to environmental stressors, including drought. In summary, rampant recombination and horizontal gene transmission decays clonality among *Streptomyces* and likely challenges phylogenetic conservation of DE and other traits among our cohort.

Despite the lack of phylogenetic signal for DE, our analyses successfully identified orthogroups strongly associating copy number with DE. Among the most enriched functions were multiple *FabG* orthogroups. This gene produces a β-ketoacyl–acyl carrier protein reductase, an essential enzyme of fatty acid synthase (FAS) II which is responsible for the first reductive step in the elongation cycle of fatty acid biosynthesis [32,57]. FAS II activity is required to produce fatty acids, the primary role of which is hydrophobicity of membrane lipids—critical barriers responsible for cellular integrity, particularly in periods of abiotic stress such as drought. *ProP*, another gene represented by multiple significant orthogroup associations is a proton symporter. This protein senses osmotic shifts and responds by importing osmolytes [28] including proline, betaine, pipecolic acid, and taurine. Notably, each of these osmolytes was recently shown to accumulate in the roots and rhizosphere of *S.bicolor* RTx430 under drought conditions [58], suggesting a potential connection between altered host metabolite production under drought and the strong colonization of *Streptomyces* isolates with greater capacity to exploit this shift in the metabolite environment.

Finally, our findings reveal a lack of consistent correlation between drought-induced microbial enrichment (DE) and plant growth-promoting rhizobacteria (PGPR) activity in sorghum, suggesting that enrichment alone may not be a reliable indicator of functional benefit to the host. Similar conclusions have been drawn in other systems, where microbial enrichment did not translate to measurable plant benefits, highlighting the need for functional validation [10,59–61]. While individual isolates belonging to the genus *Streptomyces* have often been associated with drought resilience and PGPR functions [8,62,63], our study demonstrates not only that root-associated *Streptomyces* isolates from this system did not universally exhibit these beneficial abilities, but also that those capable of benefiting their hosts are phylogenetically dispersed, and that even closely related isolates may exhibit contrasting patterns of help and harm with respect to host phenotype. This observation underscores the complexity of rhizosphere community dynamics, and suggests that microbial enrichment in the root may be governed as much by abiotic filtering, niche competition, and stochastic colonization as by targeted host recruitment. In practical terms, this means that microbiome engineering efforts aimed at improving sorghum resilience to drought must move beyond genus- or species-level classifications and instead focus on the strain-specific gene content and metabolic potential. Moreover, the presence of potentially antagonistic or neutral *Streptomyces* strains in the root microbiome may complicate inoculant design, as competitive interactions could suppress the establishment of beneficial strains. Ultimately, leveraging the full potential of *Streptomyces* in sorghum or related crops will require a systems-level understanding of microbiome composition, functional gene distribution, and environmental context. Additional studies aimed at developing this proper framework are critical for accurately interpreting microbiome responses to environmental stress and for designing effective microbiome-based interventions in crop systems.

## Materials and methods

### Field experimental design and sample collection

*Sorghum bicolor* cultivar RTx430 plants were grown in a field located at the University of California Kearney Agricultural Research and Extension (KARE) Center located in Parlier, California (36.6008°N, 119.5109°W), as described previously [58]. Sorghum seeds were sown into pre-watered plots. Starting in the third week, control treatment plants were watered for 1 h three times per week by drip irrigation (1.89 l/h flow rate). After 8 weeks, which coincided with the average onset of flowering across all plants, root samples were harvested from a subset of field samples prior to watering. All field samples were collected between 11 AM and 12 PM as described in [58]. Roots were vortexed three times for 1 min in epiphyte removal buffer (ice-cold 0.75% KH2PO4, 0.95% K2HPO4, 1% Triton X-100 in ddH2O; filter sterilized at 0.2 µm with Corning brand filters) and patted dry. All samples were immediately flash-frozen in LN2 in the field and stored at −80°C until sample processing.

### Microbial isolates

All 48 *Streptomyces* isolates used in this study originated from the same field plots at the KARE Center. All bacterial isolations were performed on the roots of sorghum cultivar RTx430 grown in this field annually over a period of three years as described in [24]. Whole genome sequencing and assembly was performed using two distinct workflows. Three isolates (SAI104, SAI190, and SAI211) were sequenced using Oxford Nanopore Technologies (ONT) v14 library prep chemistry on R10.4.1 flow cells followed by assembly with *Miniasm* v0.3 [64], *Flye* v2.9.1 [65], and polished via *Medaka* v1.8.0 (ONT 2022). All other isolates ($n = 45$) were sequenced with PacBio Sequel SMRT and assembled into draft genomes with *HGAP* v4 (0.2.1) [66]. Taxonomic identification was performed using *GTDB-tk* 2.4.0 using reference database r220 [67,68]. Phylogenetic tree construction was performed using a multiple sequence alignment of 138 known single copy genes from the Actinobacterial lineage identified with Hidden Markov Models (HMMs) using the *GToTree* v1.7.08 workflow [69–73]. Orthogroups across all genomes were identified with *Orthofinder* [33] and used to calculate core and pangenome estimates with *Micropan* [74] and subsequently visualized with *Pagoo* [75].

## Microbox experiments

To explore *Streptomyces* growth *in planta*, isolates were grown on Tryptic Soy Agar (3 g TSB powder and 15 g agar per liter) plates at 30°C for 3 days. Colonies were harvested and suspended in 1× phosphate-buffered saline (PBS) at 1 mg/ml of cell suspension. Sorghum cultivar RTx430 seeds were surface-sterilized with 10% bleach for 15 min and washed five times with sterile water. Seeds were placed on filter paper hydrated with 6 mL of sterile water in petri dishes, sealed with Micropore tape (3M) and incubated at 30°C in constant darkness for 48 hours. Germinated seedlings were inoculated in bacterial suspensions for 5 min, planted at a depth of 5 cm, inoculated again with 1 mL of bacterial slurry and covered with calcined clay (Greens Grade, Profile). Each 5 L microbox (SacO₂, Belgium) housed four seedlings planted in 1 kg of calcined clay mixed with 400 mL of 1× Hoagland's Solution. All plants were grown at 26–28°C, 16 hours:8 hours light/dark cycle, 50% humidity. For drought treatments, drought-assigned boxes were placed in a sterilized laminar flow cabinet after 1 week of growth, box lids were removed, and moisture was drawn from the boxes overnight for 16 hours. This dry-down process was repeated the following day. Water was withheld from drought-treatment boxes for the remainder of the experiment. For irrigated treatments, each plant was irrigated with 6 mL of sterile water every 7 days with no dry-down. After 5 weeks of growth, all treatments were phenotyped and harvested. Shoot fresh weight, dry weight, and % water content—((fresh weight-dry weight)/fresh weight) * 100—were measured and root tissues were harvested for microbial colonization quantification.

## Drought enrichment (DE) quantification

DNA was extracted from 35 mg of root tissue from each plant using the DNeasy PowerSoil Kit (QIAGEN, cat #47014). Isolate colonization was quantified from DNA extracts via quantitative PCR using Qiagen Quantitect SYBR Green PCR (Cat.#204143) with Actinobacterial lineage-specific 16S rRNA primers *Actino235* 5′-CGCGGCCTATCAGCTTGTTG-3′ and *Eub518* 5′-ATTACCGCGGCTGCTGG-3′ [76]. Thermal cycling used the following protocol: 95°C 15 min, and 36 3-step cycles (95°C for 15 s, 55°C for 55 s, and 72°C for 45 s) followed by a plate read; a melt curve was generated by heating from 65 to 95°C with 0.5°C increments for 5 s. Cq values from these reactions were compared to isolate DNA standard curve to interpolate 16S rRNA abundance. For each isolate, DE was calculated as such: DE = $\log_2$(Drought/Water) where "Drought" refers to the mean abundance of 4 replicates under drought and "Water" refers to mean abundance of 4 irrigated replicates. This yields a score where drought-enriched bacteria have positive scores and drought-depleted bacteria have negative scores. Phylogenetic signal for DE was tested using *phytools::phylosig* [77].

## DNA extraction, amplification, and amplicon sequencing

DNA extraction was performed using the protocol for collection of root endosphere samples using a Qiagen DNeasy Powersoil DNA extraction kit with 150 mg as starting material in the provided collection vials. The V3-V4 region of the 16S rRNA gene was PCR amplified from 25 ng of genomic DNA using dual-indexed 16S rRNA Illumina iTags 341F (5′-CCTACGGGNBGCASCAG-3′) and 785R (5′-GACTACNVGGGTATCTAATCC-3′) supplemented with PNAs [78] designed to target host-derived amplicons from chloroplast and mitochondria 16S rRNA sequences (0.75 µM of each, PNABIO, Thousand Oaks, CA). Barcoded 16S rRNA amplicons were quantified using a Qubit dsDNA HS assay kit on a Qubit 3.0 fluorometer (Invitrogen, Carlsbad, CA), pooled in equimolar concentrations, purified using Agencourt AMPure XP magnetic beads (Beckman Coulter, Indianapolis, IN), quantified using a Qubit dsDNA HS assay kit on a Qubit 3.0 fluorometer (Invitrogen), and diluted to 10 nM in 30 µl total volume before being submitted to the QB3 Vincent J. Coates Genomics Sequencing Laboratory facility at the University of California, Berkeley for sequencing using Illumina Miseq 300 bp pair-end with v3 chemistry. The full-length 16S rRNA gene (~1,500 bp) was amplified using the universal primer pair 27F (5′-AGAGTTTGATCMTGGCTCAG-3′) and 1492R (5′-TACGGYTACCTTGTTACGACTT-3′). Barcoded full-length 16S rRNA amplicons were quantified, pooled, and purified as described above for the V3-V4 region. Full-length 16S rRNA pool was finally quantified and diluted to 50 nM in 30 µL total volume before being submitted to the Department of Energy's Joint Genome Institute (DOE JGI) for sequencing using a single SMRT II cell Pacific Biosciences.

## Amplicon sequence processing and analysis for field experiment

V3-V4 16S rRNA amplicon sequencing reads were demultiplexed in QIIME2 [79] applying a minimum predicted accuracy of Q30. Chimera detection and removal were performed using DADA2 [80] and high-quality ASVs were assigned taxonomy using the August 2013 version of GreenGenes 16S rRNA gene database as described previously [58]. Circular consensus sequences (CCS) from the full-length 16S rRNA amplicons were generated from raw subreads using PacBio's SMRT Link software, applying a minimum of three full passes and a minimum predicted accuracy of Q30. Then, CCS were passed to DADA2 to generate ASVs and taxonomy was assigned also using the GreenGenes 16S rRNA gene database. All subsequent 16S statistical analyses were performed in R-v3.6.1 [81]. To account for differences in sequencing read depth across samples, samples were normalized by dividing the reads per ASV in a sample by the sum of usable reads in that sample, resulting in a table of relative abundance frequencies, which were used for analyses. Statistical significance was determined using the Holm–Sidak method, with alpha = 0.05, where each row was analyzed individually, without assuming a consistent standard deviation.

## Metabolite extraction, identification, and analysis

To evaluate the metabolomic profiles of *Streptomyces* isolates, exometabolites were collected from cultures of 12 DC strains grown in liquid tap water–yeast extract (TWYE) medium. The TWYE medium consisted of 0.25 g/L yeast extract, 0.5 g/L $K_2HPO_4$, and 18 g/L agar, prepared in tap water and supplemented with 5 mL/L of nystatin (5 mg/mL) to inhibit fungal contamination. TWYE medium was further supplemented with ~0.08 g finely ground root tissue harvested from sorghum cultivar RTx430 plants subjected to either well-watered or drought conditions. Root tissue was obtained from sorghum grown in microboxes under controlled conditions, with drought treatment applied during the final 2 weeks of growth, as described above. For each condition, 3 mL liquid TWYE cultures were inoculated with 5 μL of a $1 \times 10^5$ spores/μL suspension (four tubes per strain) and incubated at 30°C with shaking at 200 rpm for 6 days. Cells were pelleted by centrifugation in 2 mL microcentrifuge tubes. One milliliter of the clarified supernatant (spent media) was carefully transferred to a fresh 2 mL tube. This transfer step was repeated once to ensure removal of residual cells. The clarified supernatants were then snap-frozen in liquid nitrogen and lyophilized to dryness. In preparation for LC–MS, lyophilized supernatants were resuspended with 300 μl of methanol, vortexed and sonicated 10 min, centrifuged 5 min at 5,000 rpm, and supernatant centrifuge-filtered 2.5 min at 2,500 rpm (0.22 μm hydrophilic PVDF, Millipore, Ultrafree-CL GV, #UFC40GV0S), and then 150 μl was transferred to LC–MS glass autosampler vials. Internal standards used and the untargeted liquid chromatography–mass spectrometry conditions were performed as described in [58].

Metabolite identification was based on exact mass and comparison of retention time (RT) and MS/MS fragmentation spectra to those of standards run using the same chromatography and MS/MS method. A Feature-Based Molecular Networking workflow was performed using MZmine 2 and GNPS [82–84]. An MZmine workflow was used to generate a list of features (MzRT values obtained from extracted ion chromatograms containing chromatographic peaks within a narrow *m/z* range) and filtered to remove isotopes, adducts, and features without MSMS (S4 and S5 Tables). For each feature, the most intense fragmentation spectrum was uploaded to GNPS: Global Natural Products Social Molecular Networking, a web-based mass spectrometry identification tool. When a sample mass spectrum matches one deposited within the GNPS database, a putative identification was made. Totals of 3,750 polar metabolites and 15,107 non-polar metabolites were predicted including positive and negative ion modes across different treatments. Principal components analysis of metabolite profiles was performed using *prcomp::stats* function in R. All other metabolite analyses were performed using MetaboAnalyst-v4.0 [85].

## Microbial phenotyping

**Spore induction.** Spore induction in *Streptomyces* isolates was performed using MS agar, a nutrient-limited medium optimized for sporulation. MS agar was prepared by dissolving 20 g of mannitol and 20 g of soya flour in 1 L of tap water,

supplemented with 20 g of agar, and autoclaved twice at 120°C for 20 min each to enhance media clarity and reduce clumping from soya particulates. After cooling to ~55°C, the medium was poured into sterile Petri dishes and allowed to solidify. *Streptomyces* strains were streaked onto the surface of the dried plates with 200 µl of an overnight culture in TSB (30 g TSB powder per liter) and incubated at 30°C for 5–10 days. Sporulation was monitored visually by the appearance of aerial hyphae and colored, powdery spore masses.

**Siderophore detection using CAS-LB agar.** Siderophore production was assessed using a Chrome Azurol S (CAS)-LB agar assay based on the method of [86], with minor modifications. The CAS dye solution was prepared by dissolving 60 mg of CAS in 50 mL deionized water, followed by the addition of 1 mM $FeCl_3 \cdot 6H_2O$ prepared in 10 mM HCl. A solution of 91 mg Hexadecyltrimethylammonium bromide (CTAB) in 40 mL deionized water was then added slowly under constant stirring, yielding a dark blue complex. This solution was filter sterilized and stored at 4°C protected from light. LB agar medium (10 g tryptone, 5 g yeast extract, 10 g NaCl, 15 g agar per liter, pH 7.0) was prepared and autoclaved, then cooled to approximately 50°C before adding the CAS dye solution at a 1:10 ratio (e.g., 50 mL dye per 500 mL agar). The mixture was stirred gently and poured into Petri plates. *Streptomyces* isolates were spot-inoculated onto the CAS-LB plates and incubated at 28°C for 72 hours. A yellow to orange halo around colonies indicated siderophore production, resulting from iron chelation from the blue CAS–$Fe^{3+}$ complex.

**Osmotic tolerance assay.** Osmotic tolerance of microbial isolates was evaluated using sorbitol-supplemented tryptic soy broth (TSB) in both solid and liquid formats, with bacterial growth quantified via Bradford protein assay. For solid media, 10% TSB agar (3 g TSB powder per liter) was supplemented with sorbitol to final concentrations of 0.5 M, 1 M, and 1.5 M, autoclaved and poured into Petri plates. For liquid assays, TSB was similarly prepared with sorbitol concentrations of 0.5 M, 1 M, and 1.5 M, and aliquoted with 3 ml into sterile culture tubes. Bacterial isolates were pre-cultured overnight in standard TSB normalized to $1 \times 10^5$ spores and inoculated into each condition. In solid media, 5 µL of bacterial suspension was spotted onto the surface and incubated at 30°C for 48–72 h, with growth scored qualitatively. In liquid media, cultures (5 tubes per strain per condition) of 3 ml were incubated at 30°C with shaking (200 rpm) with 5 µL of bacterial suspension. One tube per condition was harvested at 0, 24, 48, 72, and 96 hours and cells were pelleted by centrifugation at 12,000 rpm for 10 min, washed once with PBS, and store at −20°C. Cells were lysed with 500 µl of 1M NaOH and total soluble protein content was measured using the Bradford assay [87]. Ten µL of lysate added to 200 µL of Bradford reagent in a 96-well plate and bovine serum albumin (BSA) was used as standard. Absorbance was read at 595 nm using a microplate reader (Infinite Nano M+, TECAN). TBS 053, *Bacillus megaterium*, isolate was used as positive control and TBS 091, *Paenibacillus lautus*, as negative control of the assays. Time-course protein quantification under increasing osmotic stress enabled the evaluation of growth dynamics and adaptation to sorbitol-induced osmotic pressure.

## Pangenome analysis

Orthogroups were identified across all 48 genomes using *OrthoFinder* 3.0.1b1 [33]. CDS for each orthogroup identified by *OrthoFinder* were aligned with *MAFFT* 7.505 [88]. *hmmbuild* and *hmmemit* from HMMER3 [89] were used to produce an HMM and consensus sequence from each multiple sequence alignment to represent the orthogroup. Finally, these cluster consensus sequences were functionally annotated with *eggNOG-mapper v2* [90,91]. Only clusters with copy number variance across all genomes >0.25 and represented in at least ¼ of the genomes ($n = 2,735$) were then subjected to phylogenetic generalized least squares analysis via *nlme::gls* (Pinheiro, Bates, and R Core Team 2025). Associations between cluster count and DE were tested while simultaneously estimating Pagel's λ *ape::corPagel* [92] to account for the effect of phylogenetic structure. In addition, a model for each gene was also fit using lambda fixed to 0—reflecting no phylogenetic signa—and 1—reflecting a Brownian motion process. Model fits were assessed by Akaike Information Criterion (AIC) to identify and select the best Pagel's λ for each orthogroup. Finally, results were filtered for standard error <0.25, slope estimate >0.25, and normality of model residuals with Shapiro–Wilk test. Remaining results were tested for functional

enrichments in Clusters of Orthologous Groups (COG) [93] relative to the background of all analyzed orthogroups ($n = 2,735$) by hypergeometric test via enrichPlot::enricher [94].

The pangenome visualization performed using Anvi'o v8 [27] used imported FASTA files, processed into a pangenome database using the snakemake pangenomics workflow [95,96]. In this workflow, open reading frames were determined using Prodigal [70], nucleotide sequences were aligned using MUSCLE [97], and amino acid sequences were aligned using DIAMOND [98]. The average nucleotide identity (ANI) was calculated using pyANI [99].

## Supporting information

**S1 Text.** Fig A. **Phenotypic and metabolomic differentiation among *Streptomyces* isolates sharing a dominant V3-V4 16S rRNA ASV. (A)** Image of the growth (on spore induction media, SIM) for all 28 *Streptomyces* isolates with 100% identity to the dominant ASV identified in the root via V3-V4 16S rRNA sequencing with original hand labeling from Fig 1A. (**B**) Ordination of non-polar exometabolomic profiling of the 12 strains following growth on liquid tap water–yeast extract (TWYE) medium showing distinct clustering of AC group 2 and 5 (replication $n = 4$). AC groups are presented by colors: AC2, purple; AC5, light blue; AC6, green mist; AC8, green. *Streptomyces* isolates ID are indicated in the corresponding plate. The data underlying this Figure can be found in https://doi.org/10.5281/zenodo.17554086. **Fig B. Ordination plot of nonpolar exometabolomics data analyzed from the spent media of all identified *Streptomyces* strains matching V3–V4 ASV in the isolate collection following growth on root tissue.** The color of each shape indicates the AC group it belongs to: AC2, purple; AC5, light blue; AC6, green mist; AC8, green. Blank control samples containing only drought root tissue (brown circles) or control irrigated root tissue (brown triangles) are shown at top right in the plot (replication $n = 4$). *Streptomyces* isolates ID are indicated in the corresponding plate. The data underlying this Figure can be found in https://doi.org/10.5281/zenodo.17554086. **Fig C. Production of representative individual siderophores putatively identified and measured by exometabolomics.** DC strains were under growth on TWYE (yellow), drought-stressed root tissue (orange), or non-stressed root tissue (blue). Cells were pelleted by centrifugation and the clarified supernatants were then snap-frozen in liquid nitrogen and lyophilized to dryness. Lyophilized supernatants were resuspended with methanol, sonicated and then transferred to LC–MS glass autosampler vials for untargeted liquid chromatography–mass spectrometry identification. The data underlying this Figure can be found in https://doi.org/10.5281/zenodo.17554086. **Fig D. Mirror match plot comparing experimental (query) and library MS/MS spectra of siderophores putatively identified by exometabolomics.** The mirror plot visualizes the spectral alignment between an experimental MS/MS spectrum (top in black) and a reference spectrum from the GNPS library (bottom in green). Shared fragment ions are indicated by aligned peaks, and spectral similarity is quantified using the cosine score. The precursor ion $m/z$, retention time (RT), ionization mode, cosine similarity, and number of matched fragment peaks are used to assess the quality and confidence of the match. **Fig E. Total siderophore production as measured by CAS-LB agar assay based on the method of Schwyn and Neilands (1987).** *Streptomyces* isolates were spot-inoculated onto the CAS-LB plates and incubated at 28°C for 72 hours. A yellow to orange halo around colonies indicated siderophore production, resulting from iron chelation from the blue CAS–$Fe^{3+}$ complex. The experiment was repeated twice with three technical replicates each. **Fig F. Osmotic tolerance assay of *Streptomyces* isolates under increasing sorbitol-induced stress.** *Streptomyces* isolates were grown in tryptic soy broth (10% TSB) supplemented with sorbitol at final concentrations of 0.5, 1.0, and 1.5 M. Bacterial growth was monitored over time (0, 24, 48, 72, and 96 hours; $n = 3$) using Bradford protein assays to quantify total biomass. This time-course approach enabled the assessment of growth dynamics and adaptation to osmotic stress. *Bacillus megaterium* (TBS 053) and *Paenibacillus lautus* (TBS 091) were included as positive and negative controls, respectively. The data underlying this Figure can be found in https://doi.org/10.5281/zenodo.17554086.
(PDF)

**S1 Table. Significant differences of ASVs identified in both full-length and V3–V4 16S rRNA sequencing datasets.** Differentially abundant ASVs were identified using normalized ASV abundance data from both full-length and V3–V4 16S

rRNA amplicon sequencing of root microbiomes. Statistical comparisons were performed using Holm–Sidak corrected significance testing ($\alpha = 0.05$).
(CSV)

**S2 Table. *In silico* predicted primer coverage based on the Silva SSU Ref NR (release 132) database compared with observed ASVs.** Predicted values indicate the proportion of reference sequences matching each primer set, while observed values represent the proportion of ASVs recovered in this study.
(CSV)

**S3 Table. PERMANOVA analysis of the phenotyping data collected from the osmotolerance and siderophore production assays.** The analysis was conducted using the Euclidean distance of control and DC isolate samples per phenotyping condition.
(CSV)

**S4 Table. Library-based spectral matches (GNPS DB_results).** This table summarizes high-confidence spectral matches between experimental MS/MS spectra and reference spectra in the GNPS library. These results support the identification of known metabolites present in the sample.
(XLSX)

**S5 Table. Summary of molecular clusters generated by GNPS molecular networking.** This table provides information on the consensus MS/MS clusters identified through molecular networking analysis using GNPS. Each cluster represents a group of MS/MS spectra with similar fragmentation patterns, corresponding to a putative unique metabolite or structurally related compounds. The table includes the cluster index, consensus precursor $m/z$, retention time, number of spectra per cluster, and the number of unique sample files contributing to each cluster.
(XLSX)

## Acknowledgments

We thank the staff of the Kearney Agricultural Research Center for their help in sample collection and field preparation.

## Author contributions

**Conceptualization:** Citlali Fonseca-Garcia, Dean Pettinga, Daniel Caddell, Matthew F. Traxler, Devin Coleman-Derr.

**Data curation:** Citlali Fonseca-Garcia, Dean Pettinga, Katherine Louie, Benjamin P. Bowen, Trent R. Northen.

**Formal analysis:** Citlali Fonseca-Garcia, Dean Pettinga, Hannah Ploemacher, Katherine Louie, Benjamin P. Bowen, Alen Zimic-Sheen, Trent R. Northen, Devin Coleman-Derr.

**Funding acquisition:** Matthew F. Traxler, Devin Coleman-Derr.

**Investigation:** Citlali Fonseca-Garcia, Dean Pettinga, Devin Coleman-Derr.

**Methodology:** Citlali Fonseca-Garcia, Dean Pettinga, Daniel Caddell, Katherine Louie, Benjamin P. Bowen, Joelle Park, Jesus Sanchez, Trent R. Northen.

**Supervision:** Devin Coleman-Derr.

**Visualization:** Citlali Fonseca-Garcia, Dean Pettinga, Hannah Ploemacher.

**Writing – original draft:** Citlali Fonseca-Garcia, Dean Pettinga, Hannah Ploemacher, Devin Coleman-Derr.

**Writing – review & editing:** Citlali Fonseca-Garcia, Dean Pettinga, Daniel Caddell, Hannah Ploemacher, Katherine Louie, Benjamin P. Bowen, Joelle Park, Jesus Sanchez, Alen Zimic-Sheen, Matthew F. Traxler, Trent R. Northen, Devin Coleman-Derr.

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
