## [Editor Report · Decision Letter 0]

7 Jul 2025

Dear Dr Coleman-Derr,

Thank you for submitting your manuscript entitled "Uncovering the hidden diversity and functional roles of root endophytic Streptomyces under drought stress" for consideration as a Research Article by PLOS Biology. My sincere apologies for taking this long.

Your manuscript has now been evaluated by the PLOS Biology editorial staff, as well as by an academic editor with relevant expertise, and I am writing to let you know that we would like to send your submission out for external peer review.

Once your full submission is complete, your paper will undergo a series of checks in preparation for peer review. After your manuscript has passed the checks it will be sent out for review. To provide the metadata for your submission, please Login to Editorial Manager (https://www.editorialmanager.com/pbiology) within two working days, i.e. by Jul 09 2025 11:59PM.

Kind regards,

Melissa

Melissa Vazquez Hernandez, Ph.D.

Associate Editor

PLOS Biology

---

## [Decision Letter · Decision Letter 1]

2 Sep 2025

Dear Dr Coleman-Derr,

Thank you for your patience while your manuscript "Uncovering the hidden diversity and functional roles of root endophytic Streptomyces under drought stress" was peer-reviewed at PLOS Biology. It has now been evaluated by the PLOS Biology editors, an Academic Editor with relevant expertise, and by three independent reviewers. Please accept my sincere apologies for the delay.

In light of the reviews, which you will find at the end of this email, we would like to invite you to revise the work to thoroughly address the reviewers' reports. As you will see below, majority of reviewers are positive about the relevance and novelty of the study, yet some concerns have raised during revision. Reviewer 1 has several questions like the differences in taxonomic composition (which may require validation) and if analysing the linkage between exametabolomic molecules and DE/PGP scores might be more direct evidence. Reviewer 2 has several comments on how the study is written and organized. Reviewer 3 suggests to expand the analysis on gene neighbourhoods with a focus on FabG, as well as on the impact of monoassociation on the plant. We agree with all reviewer concerns and would require some additional experimental revisions to address them, as we consider that this would strengthen the work.

IMPORTANT: after discussion with the Academic Editor and the reviewers, we would like to emphasize point 2 of reviewer 3 as this will give a focus on the plant physiological responses to drought.

Given the extent of revision needed, we cannot make a decision about publication until we have seen the revised manuscript and your response to the reviewers' comments. Your revised manuscript is likely to be sent for further evaluation by all or a subset of the reviewers.

**IMPORTANT - SUBMITTING YOUR REVISION**

*Re-submission Checklist*

*Published Peer Review*

*PLOS Data Policy*

*Blot and Gel Data Policy*

Sincerely,

Melissa

Melissa Vazquez Hernandez, Ph.D.

Associate Editor

PLOS Biology

REVIEWERS' COMMENTS

Reviewer #1 (Yi Song):

Drought is one of the most serious abiotic stresses for agricultural production worldwide, while we have very limited biotechnologies to combat drought relative to pathogens. Accumulating evidence suggest that plant microbiome changes contribute to plant drought tolerance, however, the microbial ecologic and genetic mechanisms underlying drought induced microbiome changes are still not clear. The authors tried to conduct in-depth mechanistic exploration for their previous found interesting phenomenon: the genus Streptomyces is consistently enriched in the roots of drought-stressed plants and confer drought protecting activity for crops. One of the really cool point of this manuscript is that they carefully combined sequencing based microbiome analyses and isolates based strain functional and genomic analyses. This basically is a perfect "binning" process similar to metagenomics based microbe genome assembly, but definitely much higher accuracy because they successfully isolated and purified representative strains. They further explored the physiological and genomic correlations with their drought enrichment folds, and found that there is not strong correlation between drought enrichment folds and potential drought protection ability. This is of significance to guide the characterization and application of beneficial microbiome in agriculture. Overall the MS is rigorously designed and clearly written.

I have a couple questions related to the manuscript:

[1] One of my biggest surprise is that there are huge differences in the taxonomic composition between full-length and V3-V4 region based 16S rRNA gene sequencing results (1% Actinobacteria in full-length control group, while around 30% in v3-v4 approach?). This part lacks potential explanation or cross validations? Are there any similar comparison study in animal gut or environmental systems? Is this common to observe such a drastic (or totally different) differences between those two approaches?

[2] I would be a bit more careful to draw the conclusion that "this challenges the conventional 'cry for hlep' framework." Because the "cry for help" concept itself is not a strict "mathematical law", in contrast, it is more like an "ecological trend" which usually helpful to guide the predication and discovery of potential beneficial microbes at the family or genus level. It helps researchers to narrow down potential "target" microbes for functional validations. Moreover, the "cry for help" phenomenon seems to be supported by a lot genetic and metabolic evidence from host side. Meanwhile, the input from this study (DE) was got from mono-association system and Quantitative profiling, which is very different from the relative abundance and natural soil community (which was usually the input for "cry for help" studies). Recent study also indicate the importance and relevance for relative fold changes for predicting core drought responsive microbes [ PMC12043206]. I guess simply delete line 439-441 could address this issue.

[3] The authors successfully isolated diverse Streptomyces strains and performed exometabolomic profiling studies, which is cool. However, the authors only analyzed the correlation between genomic loci and DE scores or growth promoting effect in the study. I am wondering would that be more direct to analyze the linkage between exometabolomic molecules and DE/PGP scores? Would that be more direct to reveal potential rules of small molecules controlling positive DE scores in Streptomyces strains.

[4] fig3 seems hard to understand. Is it lacking scale bar? Is there an easier way to illustrate this, or could we separate them into more clear different graphs?

Reviewer #2:

This manuscript presents an observational study on the genetic diversity of in bacterial family of Streptomyces often associated with plants under drought stress. The study uses well-established genomics methods to analyze diversity in the Streptomyces strains present in sorghum roots under drought stress. The results are not very surprising in that it is well known that related microbial strains can have vasty different functionality. The novelty of the study seems to be in linking drought enrichment with possible growth promotion and showing that there is no link. The manuscript, while relatively well written in the technical sense is hard to follow. The results are presented as a mixture of hypotheses and methods and it remains unclear whether the main outcome of this study was the comparison between the two sequencing methods or the observations of the variability in functionality of the strains. The abstract and introduction emphasize the patter while the methods and discussion emphasize the former. The emphasis on the sequencing techniques comes as a surprise to the reader because this aspect of the study is not mentioned in the introduction at all. I suggest reorganizing the manuscript around one focus area (the one that the authors think has the highest impact/importance to the community) and clearly introducing the background of that including information about hypotheses and the methods to test the hypotheses. In its current form, the introduction seems somewhat superficial about what is known about the genus Streptomyces. The discussion also revolves around methodology rather than the deeper meaning of the results. If the study focus is the diversity in functionality of Streptomyces strains, then discussing that and its implications for plant or microbiome performance would strengthen the manuscript.

Specific comments:

-Terminology: Please define DE clearly in the manuscript and indicate when the text refers to the calculated DE value and when the word "enrichment" is used to refer to presence or absence of Streptomyces in the roots. Now these two terms are somewhat mixed making the manuscript hard to follow.

-Tenses: Please use the same tense consistently to present what was done in this study. Most journal like the style of everything done in this study presented using past tense. Now the methods and results jump between past and present tense.

Figures: Why is the phylogenic tree needed in Fig. 3? It is partially covered. If it is important, why is it not shown fully? Also the results refer quiet extensively to the figures in the supplementary materials. Please consider bringing the key figures that show results to the main text as this journal does not have a figure limitation.

-Line 49: This seems like an overstatement as this study addresses only a limited number of strains in the genus.

-Line 77: It is unclear what "this response" refers to here. Please clarify.

-Line 81-83: If the focus of the study is revealing functional diversity, please explain what aspect of that these methods can reveal.

-Line 104-107: Here the use of the word "enrichment" is confusing. Please clarify what is the metric of enrichment here.

-Line 114-115: Please address the statistical significance of these differences. is 16.7% significantly larger than 12.3%

-Line 175: Please add "that" between suggest and metabolite.

-Line 229-231: Why is the word "despite" the right word to use in this sentence? I have hard time seeing any contradiction between the statements in the first and second part of the sentence.

-Line 232: Did this group include the preciously used strains or where these strains completely new?

-Line 305-306: This has been already told in the results, and it is material that could be only in the methods.

Reviewer #3:

This manuscript clearly and logically describes an important set of studies that demonstrate the amplicon sequencing results do not necessarily connect to critical microbial functions/activities in the study system. In this case, the authors investigate a well-known findings that bacteria in the genus Streptomycs increase in abundance when plants experience drought conditions. This manuscript makes important contributions to characterize which sequence variants for a V3-V4, as well as a full length 16S rRNA gene sequence to provide additional resolution. Most importantly, this study connects amplicon sequencing data to a collection of isolates to test several contributions to plant interactions, including under drought conditions. Overall, the manuscript is compelling and presents important contributions to the field. There are a couple of small clarifications and connections that could further increase the impact of these findings, particularly for the data presented in Figures 4 and 5.

1) Figure 4 shows gene copy number for FabG and ProP. Lines 290-293 mentions 8 gene clusters annotated as FabG. Does this mean there are 8 different types of gene clusters? Do all of the genes annotated for both FabG and ProP have conserved genes surrounding it? Additional description of the gene neighborhoods for each of these genes could nicely highlight potential explanations for the results described in this paragraph.

2) Figure 5 presents data from an important experiment and the conclusion has important implications for how the field should interpret differential abundance data in the future. However, biomass as the metric to make these conclusions is a bit removed from the actual plant physiological responses to drought. What is the impact of monoassociation on plant photosynthetic measurements, stomatal. conductance, or relative water content in leaves? Any of these measurements, particularly on interesting comparisons highlighted already by the authors (e.g. DC01, SAI126, etc.) would provide this manuscript with an important foundation for future studies.

---

## [Editor Report · Decision Letter 2]

27 Oct 2025

Dear David,

Thank you for your patience while we considered your revised manuscript "Uncovering the hidden diversity and functional roles of root endophytic Streptomyces under drought stress" for publication as a Research Article at PLOS Biology. This revised version of your manuscript has been evaluated by the PLOS Biology editors and the Academic Editor.

Based on our Academic Editor's assessment of your revision, we are likely to accept this manuscript for publication, provided you satisfactorily address the remaining editorial points. Please also make sure to address the following data and other policy-related requests.

1) We routinely suggest changes to titles to ensure maximum accessibility for a broad, non-specialist readership, and to ensure they reflect the contents of the paper. In this case, we would suggest a minor edit to the title, as follows. Please ensure you change both the manuscript file and the online submission system, as they need to match for final acceptance:

"Enrichment of root-associated Streptomyces strains in response to drought is driven by diverse functional traits and does not predict beneficial effects on plant growth"

2) The current version is missing Figures 1 and 2. Please provide them.

Please supply the numerical values either in the a supplementary file or as a permanent DOI’d deposition for the following figures:

Figure 3, 4A-D, 5A-D, 6, Supplemenary figures A, B, C, F (and if necessary also for Figs 1 and 2)

4) Please cite the location of the data clearly in all relevant main and supplementary Figure legends, e.g. “The data underlying this Figure can be found in S1 Data” or “The data underlying this Figure can be found in https://doi.org/10.5281/zenodo.XXXXX”

5) Supplementary files (e.g., excel). Please ensure that all data files are uploaded as 'Supporting Information' and are invariably referred to (in the manuscript, figure legends, and the Description field when uploading your files) using the following format verbatim: S1 Data, S2 Data, etc. Multiple panels of a single or even several figures can be included as multiple sheets in one excel file that is saved using exactly the following convention: S1_Data.xlsx (using an underscore).

6) Many thanks for supplying your data/code in Github; however, because Github depositions can be readily changed or deleted, please make a permanent DOI’d copy (e.g. in Zenodo) and provide this URL. I am aware that you also provide a link to Zenodo but I do not have access to see the information provided there.

7) Please provide the tree files for the phylogenetic trees in Figures .1B, 4A, 6 Please make sure all relevant figures have scale bars.

8) Please ensure that your Data Statement in the submission system accurately describes where your data can be found and is in final format, as it will be published as written there. Please make sure that all links are accesible.

9) Thank you for providing the underlying code in GitHub. However, because Github depositions can be readily changed or deleted, please make a permanent DOI’d copy (e.g. in Zenodo) and provide this URL in the manuscript and Data Availability Statement.

We expect to receive your revised manuscript within two weeks.

*Published Peer Review History*

*Press*

Sincerely,

Melissa

Melissa Vazquez Hernandez, Ph.D.

Associate Editor

PLOS Biology

---

## [Editor Report · Decision Letter 3]

12 Nov 2025

Dear Devin,

Thank you for the submission of your revised Research Article "Enrichment of root-associated Streptomyces strains in response to drought is driven by diverse functional traits and does not predict beneficial effects on plant growth" for publication in PLOS Biology. On behalf of my colleagues and the Academic Editor, Cara Helene Haney, I am pleased to say that we can in principle accept your manuscript for publication, provided you address any remaining formatting and reporting issues. These will be detailed in an email you should receive within 2-3 business days from our colleagues in the journal operations team; no action is required from you until then. Please note that we will not be able to formally accept your manuscript and schedule it for publication until you have completed any requested changes.

PRESS

Sincerely, 

Melissa

Melissa Vazquez Hernandez, Ph.D., Ph.D.

Associate Editor

PLOS Biology
